# Evaluation of Survey Nonresponse in Measuring Cardiometabolic Health Risk Factors and Outcomes among Sexual Minority Populations: A National Data Linkage Analysis

**DOI:** 10.3390/ijerph20075346

**Published:** 2023-03-31

**Authors:** Neeru Gupta, Samuel R. Cookson

**Affiliations:** Department of Sociology, University of New Brunswick, Fredericton, NB E3B 5A3, Canada

**Keywords:** sexual orientation, sexual minorities, health surveys, data linkage, health measurement

## Abstract

Understanding cardiometabolic health among lesbian, gay, and bisexual (LGB) people is challenged by methodological constraints, as most studies are either based on nonprobability samples or assume that missing values in population-based samples occur at random. Linking multiple years of nationally representative surveys, hospital records, and geocoded data, we analyzed selection biases and health disparities by self-identified sexual orientation in Canada. The results from 202,560 survey respondents of working age identified 2.6% as LGB, 96.4% as heterosexual, and <1.0% with nonresponse to the sexual identity question. Those who did not disclose their sexual identity were older, less highly educated, less often working for pay, and less often residing in rural and remote communities; they also had a diagnosed cardiometabolic condition or experienced a cardiometabolic-related hospitalization more often. Among those reporting their sexual identity, LGB individuals were younger, more likely to smoke tobacco or drink alcohol regularly, more likely to have heart disease, and less likely to have a regular medical provider than heterosexual persons. This investigation highlighted the potential of leveraging linked population datasets to advance measurements of sexual minority health disparities. Our findings indicated that population health survey questions on sexual identity are not generally problematic, but cautioned that those who prefer not to state their sexual identity should neither be routinely omitted from analysis nor assumed to have been randomly distributed.

## 1. Introduction

The use of record linkages across different types of datasets is becoming increasingly important in advancing the measurement and understanding of socio-spatial inequalities in health, especially among minority or hard-to-sample populations [1,2,3,4]. In several high-income countries, studies using representative surveys or administrative data alone (unlinked) have demonstrated disparities in mental health, psychiatric morbidity, and problematic substance use among sexual minorities, including lesbian, gay, and bisexual (LGB) adults [5,6,7,8,9,10,11]. A smaller but growing body of literature has focused on inequalities in cardiometabolic health risk factors and outcomes, including a heightened risk of diabetes mellitus among sexual minorities compared with their heterosexual peers [6,12,13,14]. Some studies have provided information on disparities in mental health and healthcare experiences among rural LGB persons, although many lack urban comparators [15]. The combination of conventional social science data sources (e.g., from random sample surveys) with routinely collected healthcare information (e.g., patient records) is opening up new research pathways for understanding social inequalities in health, but the process of using these data for population analytics is not without its challenges [16,17]. Many of the limitations of record linkages, notably with regard to the potential biases arising from specific missing attribute values, have not been well documented in scientific publications [16].

A certain degree of nonresponse in survey data collection is common and may introduce measurement bias; however, a recent review of the literature found that the evidence was inconclusive on the magnitude or direction of any such bias [18]. Specifically, regarding sexual minority populations, studies in Canada and the United Kingdom have described differing sociodemographic characteristics among survey respondents who did not disclose their sexual orientation from self-identified LGB and heterosexual respondents [7,19], whereas in the United States, nonresponse rates appear to be declining and converging over time across groups [20]. Previous studies have focused on demographic and linguistic factors in sexual orientation nonresponse, drawing on unique (unlinked) sources. Evidence is notably scarce on nonresponse in the self-identification of sexual orientation for use in data linkage analyses. It is often assumed that missing microdata values in population-based data occur at random [16]. Our interest was to statistically test whether missing values for the survey question on sexual identity were dependent upon the characteristics of individuals, and thus themselves offered meanings that should be considered in understanding sexual minority health disparities.

This novel study leverages nationally-representative linkable survey, administrative, and geospatial datasets to examine the associations between cardiometabolic risk factors and healthcare outcomes with sexual orientation self-identification among working-age adults in Canada. The objectives are three-fold: first, to assess the differences by response status to a survey question on sexual identity in terms of cardiometabolic health risk factors; second, to assess the differences in terms of respondents’ place of residence across the rural–urban continuum; and third, to assess the differences in terms of cardiometabolic conditions, access to primary care services, and cardiometabolic-related hospitalization. The use of multiple types of datasets will advance the measurement and understanding of sexual minority health across otherwise underexplored variables.

## 2. Materials and Methods

### 2.1. Data Sources and Target Population

Data on sexual identity were drawn from the Canadian Community Health Survey (CCHS), a large-scale household survey that collects information related to health determinants, health status, and healthcare utilization. Conducted annually by Statistics Canada, the CCHS was the first national survey in the country to include a question on sexual orientation. The sampling coverage excludes a few groups (e.g., persons residing on Indigenous reserves, full-time members of the Canadian Forces, and persons living in an institutional collective dwelling such as a nursing home), altogether representing less than 3% of the population [21]. The present analysis used information from those respondents who consented to have their data shared and linked with certain partner sources (>90% of the original CCHS respondents); to compensate for the loss of some respondents, a share weight adjustment was applied to the microdata to maintain representation [21]. To mitigate the potential impact of survivorship bias among older adults in observational studies on cardiometabolic health using household survey data [22], we limited the analyses to the working-age population (aged 18 to 59 years).

Given the stringent privacy and confidentiality protocols governing the CCHS, ten years of survey data (2008–2017) were pooled together to obtain sufficient sample sizes of LGB persons. Individuals’ sexual identity was ascertained by the question “Do you consider yourself to be...”, with three response options categorized behaviourally (including interviewer prompts): “heterosexual? (sexual relations with people of the opposite sex)”; “homosexual, that is lesbian or gay? (sexual relations with people of your own sex)”; or “bisexual? (sexual relations with people of both sexes)”. Most survey questions allowed the interviewer to record options of “don’t know” and “refusal” (not read aloud to respondents). We considered nonresponse to the sexual identity question as encompassing either of the last two options (i.e., “don’t know” or “refusal”).

To advance our understanding of the factors associated with sexual identity, we linked the CCHS data to ten years of routinely collected administrative hospital data from the 2008/09–2017/18 Discharge Abstract Database (DAD) and to the geocoded Index of Remoteness. Given Canada’s universal healthcare coverage system, the nationally standardized DAD is deemed to capture a virtually complete recording of inpatient stays for 12 of the country’s 13 jurisdictions (excluding facilities in the province of Quebec, which record hospital morbidity differently) [23]. The CCHS–DAD linkage was based on a probabilistic record-matching process developed at Statistics Canada, described elsewhere [24,25]. These data were then linked by individuals’ residential postal code to the Index of Remoteness, a statistical measure of the geographic proximity to population centres and travel radius to key socioeconomic and health services for all 5125 populated communities (census subdivisions) in the country [26].

### 2.2. Measures

To examine the differences in self-identified sexual orientation in terms of cardiometabolic risk profiles and outcomes, we used a range of CCHS data on individuals’ sociodemographic, behavioural, and health status characteristics. Sociodemographic variables included sex (whether the respondent categorized themselves as “male” or as “female”), age group, educational attainment, marital status (whether or not the respondent was married or in a common-law union, with a partner of the opposite or same sex), and employment status at the time of the survey. Cardiometabolic risk factors included body mass index class (whether or not the respondent’s body height and weight classified them as overweight or obese), tobacco use (whether or not the person presently smokes cigarettes at least occasionally), and alcohol consumption (whether or not the person regularly drinks alcoholic beverages) [21]. Health status indicators were based on respondents’ answers to questions on whether or not they had ever been told by a health professional they had diabetes (any type), hypertension, or heart disease, or were suffering from the effects of a stroke [21].

To assess the differences according to respondents’ place of residence, we ranked the continuous Index of Remoteness into quintiles, to broadly distinguish communities as highly accessible, accessible, moderately accessible, remote, or very remote. Highly accessible communities (i.e., quintile 1) essentially represent the largest metropolitan agglomerations of the country and the areas closest to them, which have facilitated access to a broad range of services for social and economic development. Very remote communities (quintile 5) are those generally characterized by sparsely distributed populations and limited transportation infrastructures [27].

We further assessed the differences in self-identified sexual orientation considering two measures of healthcare use. The first, captured in the CCHS as an important tracer for chronic disease prevention and management, was whether or not the respondent reported having a regular healthcare provider, defined as a health professional regularly seen or talked to when needing care or advice about health. The second, measurable only following the linkages across the survey and administrative sources, was whether or not the individual was hospitalized at least once over the period of observation for selected cardiometabolic conditions. We included inpatient stays for a number of ambulatory care-sensitive conditions—diabetes (types 1 or 2), hypertension, cardiac arrhythmia, health disease, heart failure, or stroke—that is, conditions for which hospitalization is expected to be largely avoidable with access to high-quality primary care [28,29].

### 2.3. Data Analyses

A quality evaluation was diagrammed across the steps of the dataset linkages, distinguishing four categories of missing observations: those excluded due to the absence of certain subpopulations in any of the original datasets (e.g., hospital records in the DAD for residents of Quebec); those excluded due to duplicate records within a given database (e.g., hospital readmissions for the same individual over the period of observation); those excluded due to errors in certain fields that hindered the dataset linkage process (e.g., invalid postal code); and those excluded due to a missing value for specific individuals (e.g., because a respondent to the CCHS chose not to disclose certain information). We focused on completed versus missing responses to the survey question on sexual orientation.

Chi-square testing was used to differentiate cardiometabolic risk factors, health outcomes, relative remoteness, and healthcare use indicators by sexual identity disclosure. Two rounds of statistical tests were conducted: (1) a first round comparing those who did not respond categorically to the survey question on sexual identity versus those who provided valid information, and (2) a second round among those with valid responses, comparing those who self-identified as LGB versus those belonging to the heterosexual majority.

Bootstrapped sampling weights were applied to the linked data to ensure the population representation of the results and robustness of the test statistics (with the level of significance set at 0.05). The analyses were conducted using the Stata statistical software program [30]. The de-identified datasets were accessed in the secure facilities of the New Brunswick Research Data Centre, located at the University of New Brunswick. All (unweighted) sample and (weighted) population counts were rounded and vetted to meet Statistics Canada data privacy and disclosure protocols.

## 3. Results

### 3.1. Population Selection

Figure 1 presents a data-flow diagram of the record linkages across the three types of sources. Of the 499,600 survey respondents who consented to have their data linked with other administrative health datasets, 279,020 (55.8%) were excluded due to falling outside the target subpopulation for this study (i.e., aged 18–59 years and residing in any of the eligible jurisdictions). Another 14,960 (3.0%) were excluded due to a missing value for any of the survey-based predictor measures, and 3080 (0.6%) due to an invalid linkage field value. The final sample thus counted 202,540 respondents of working age. Upon weighting, the sample was expected to represent 14,328,500 person-years of exposure to self-reporting on sexual orientation (i.e., heterosexual, LGB, or nonresponse).

### 3.2. Sexual Identity Nonresponse

The survey data for the population aged 18–59 identified 2.66% as LGB, 96.37% as heterosexual, and 0.97% as nonresponse to their sexual identity (Table 1). Compared with those who provided valid sexual identity information, respondents who did not answer were more often female (overrepresented at 1.1% of the target population; *p* < 0.05), in the older age range of 45–59 years, having at most a secondary level of education, and not currently working. Those not answering were less often overweight or having obesity, tobacco smokers, or regular alcohol drinkers.

Compared with those who answered the survey question on sexual orientation, nonresponders more often resided in the most urbanized and accessible parts of the country and, conversely, less often in more rural and remote areas (*p* < 0.05) (Table 2). Nonresponders were significantly more likely to report having diabetes, hypertension, heart disease, or the effects of a stroke (Table 3). They were hospitalized more often for a cardiometabolic condition versus those who declared their sexual identity, although the two groups were as likely to have a regular primary care provider.

### 3.3. Sexual Minority versus Heterosexual Identity

Among those reporting their sexual orientation in the survey, LGB individuals identified significantly more often as female or in the younger age group of 18–29 years compared with their heterosexual counterparts (Table 1). LGB individuals reported less often cohabiting with a partner or working for pay. They were over-represented among tobacco smokers and regular alcohol drinkers.

LGB individuals more often resided in the most urbanized communities of the country compared with heterosexual adults (Table 2). They were significantly more likely to have been diagnosed with heart disease but not other cardiometabolic conditions (Table 3). They were less likely to have a regular medical provider than heterosexuals.

## 4. Discussion

This novel study linking multiple types of national data sources revealed non-negligible information biases in identifying lesbian, gay, bisexual, and other sexual minority groups in population-based data collection. While it is increasingly expected that health surveillance tools capture information on sexual minority status as a social determinant of health, small proportions of LGB individuals in relation to heterosexual persons restrict empirical investigations [31]. We used one of the largest known samples of LGB working-age adults by pooling ten years of comparable survey data (n = 5400 persons identifying as LGB and n = 195,420 persons with heterosexual identity). Those who did not disclose their sexual orientation remained relatively very few (n = 1720 or less than 1% with survey nonresponse), yet exhibited some important differences in sociodemographic characteristics, geographical distribution, health-promoting risk factors, cardiometabolic conditions, and cardiometabolic-related hospitalization incidence compared with those who did answer this survey question.

Our study was consistent with findings elsewhere indicating that survey questions on sexual orientation are not controversial in contexts where the human rights of LGB persons are protected and promoted; at the same time, social patterning in the nonresponse rate suggests those who prefer not to state their sexual identity should neither be simply excluded from analysis nor assumed to have been randomly distributed [7,19,32]. The present results are unique in that we compared the different sexual identity groups by cardiometabolic health risks and healthcare outcomes. In Canada and elsewhere, population data linkage studies are fostering valuable insights on the socioenvironmental determinants of health and healthcare use, although only a narrow subset of these have focused on cardiometabolic health disparities among sexual minority populations [1,13]. We found important unmet needs for primary care services among sexual minority groups, despite Canada’s universal healthcare system. Observations of poorer cardiometabolic outcomes among sexual minorities are being increasingly attributed to chronic stress associated with belonging to a socially marginalized and stigmatized group [33,34,35]. Identity concealment, or hiding one’s sexual orientation, may be confounded by stigma and exacerbate associated stress processes and barriers to quality healthcare [32,36].

We further investigated the intersections of rurality, LGB identity, and survey nonresponse. Although it has been suggested that people in rural communities may be less inclined to disclose their sexual identity, research recognizing the diversity of sexual minority health across rural landscapes has been limited [37]. We found that, in the Canadian context, residents of more rural and remote parts of the country were significantly less likely to identify as LGB than those residing in more urbanized areas; however, they were more likely to report their sexual identity in a survey. Such results reinforce the need for rural healthcare services to address not only stigma-related health disparities among LGB persons but also the perceived challenges associated with identity disclosure [37].

### Study Limitations

As with all kinds of observational research involving hard-to-reach populations, certain limitations to this study are noted. First, it is likely that assessing sexual orientation through a single survey question focused on identity measurement may underestimate sexual minority populations, who reflect a spectrum of identities and experiences. Respondents may not disclose for various reasons, such as not having previously considered the issue, being unsure of their orientation, not understanding the question, or not associating their identity with the provided labels, each of which in turn may be associated with sociodemographic patterning [19,32]. A previous evaluation of the CCHS approaches to data collection among adults suggested that identity measures may perform better than measures of sexual behaviours and attractions [38]. However, culturally specific sexual identity terms, such as Two-Spirit or same-gender-loving, may be overlooked in national databases. Information gaps on Indigenous people persist in the Canadian health research ecosystem and are particularly acute at the intersection of Two-Spirit health in rural and reserve communities [39].

Second, despite the pooling and linking of multiple years of data, the sample size of cardiometabolic-related hospitalizations remained small (n = 300 first admissions among LGB persons). It is possible that a lack of significant findings in some cases signalled low statistical power, rather than the likelihood that the results signified a true effect. An analysis elsewhere of the sexual orientation question in the CCHS using multiple years of pooled (unlinked) survey data was required to suppress the public dissemination of certain disaggregated information (such as by Indigenous, Black, or other ethnocultural backgrounds) due to failing to meet Statistics Canada’s minimum criteria for privacy protection [19]. The nature of any differences in the characteristics of LGB persons who consented to have their survey data linked with administrative sources compared with those who did not consent, and thus were not profiled in the present analysis, remains unknown.

Third, the application of survey sampling weights, designed to ensure representativity of the general household population for a given period, depends on the analytical goals at hand [40], and for which the implications for studying specific domains in detail have not been validated. Given Canada’s vast territory, the risk of sampling errors may be increased for analyses among smaller, geographically dispersed groups (e.g., LGB persons) and for less populated regions (e.g., more rural and remote areas). Oversampling or respondent-driven sampling of small subpopulations could help enhance future empirical investigations of health and healthcare inequalities [31,41].

## 5. Conclusions

This study represents the first (to our knowledge) national-level investigation leveraging linkable population-based datasets to comprehensively assess sexual minority cardiometabolic risk profiles, health disparities, and nonresponse bias. The digital revolution has led to an explosion of population health data, but the integrity of their use for selecting subsamples of hard-to-reach minority groups has been under-evaluated. Moreover, much of the public health literature on sexual minorities has focused on issues of sexual health, mental health, and behavioural disorders. Our aim was to evaluate the challenges and feasibility of using national population datasets to advance the measurement of chronic physical health and healthcare metrics among (small) LGB subpopulations. Based on data from Canada, relying on self-identified information on sexual orientation was not generally found to be problematic, although a cautionary tale emerged in terms of the nonrandom distribution of nonresponses according to selected risk factors, geographies, and cardiometabolic conditions that should be considered in socioepidemiological research.

The reasons for an individual to withhold sensitive information in a survey or clinical setting may be rooted in social, cultural, and personal aspects, which may vary over the life span. Future investigations should ensure respondents are clearly informed on why such information is being collected and the trustworthiness of the collecting institution in protecting the privacy of their data; moreover, the response categories used in coding sexual orientation quantitatively should not necessarily be assumed to be stable over time and across collections [16]. At the time of this study, for example, the Canadian statistical agency had initiated testing new mapping standards for sexual identities beyond the default three categories (heterosexual, gay or lesbian, or bisexual) to classify other minority statuses (e.g., pansexual, asexual, queer, Two-Spirit) with inclusive definitions that could potentially be considered for meaningful comparisons [42]. More qualitative research is also needed for enhanced psychometric evaluation of why certain individuals opt not to disclose their sexual identity in national surveys to inform equity-promoting health programs and services.

## Figures and Tables

**Figure 1 ijerph-20-05346-f001:**
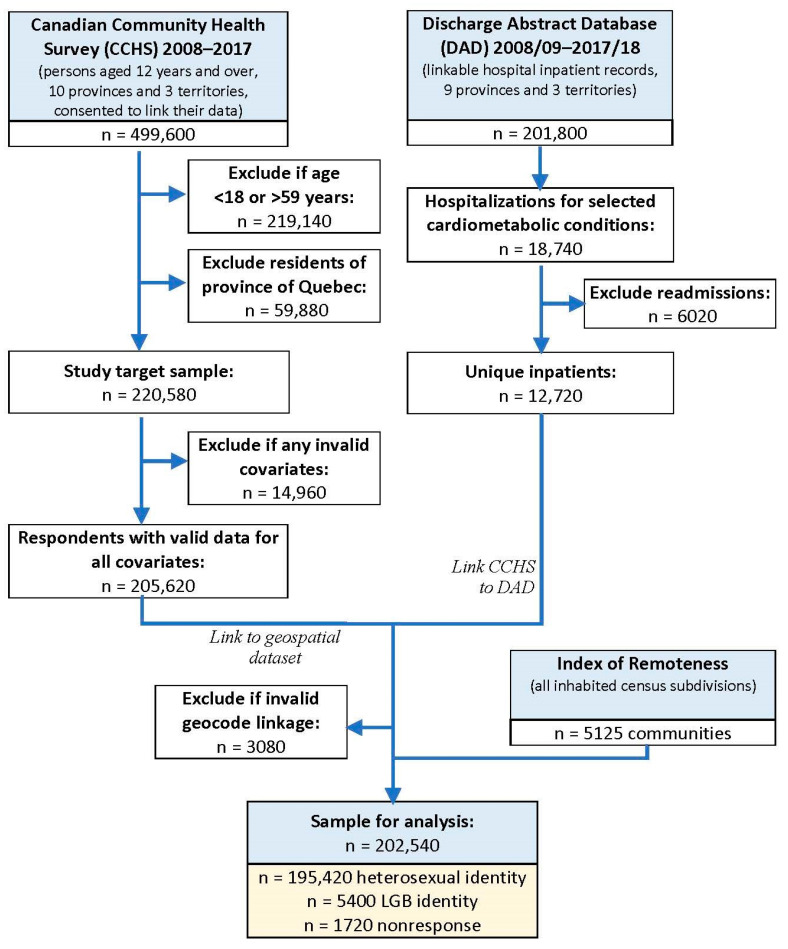
Flowchart of data linkage for the study population.

**Table 1 ijerph-20-05346-t001:** Percentage distribution of survey respondents aged 18–59 by their response to the question on sexual identity, according to sociodemographic characteristics and cardiometabolic risk factors.

Characteristic		(1) Did Not Answer	(2) Answered	
			Lesbian, Gay, or Bisexual	Heterosexual
Sex	Female	1.1 *	2.9 *	96.0
	Male	0.8 *	2.4 *	96.8
Age group	18–29 years	0.8 *	4.3 *	94.9
	30–44 years	0.9 *	2.3 *	96.8
	45–59 years	1.1 *	1.8 *	97.1
Educational attainment	At most secondary school	1.3 *	2.7	96.0
	Any postsecondary education	0.8 *	2.7	96.5
Marital status	Married or common-law	0.9 *	1.4 *	97.7
	Not in union	1.1 *	4.6 *	94.3
Employment status	Worked for pay	0.8 *	2.5 *	96.7
	Did not work	1.6 *	3.4 *	95.0
BMI class	Overweight or obese	0.9 *	2.4 *	96.7
	Not overweight	1.0 *	3.0 *	96.0
Tobacco smoker	Smokes daily or occasionally	0.7 *	3.7 *	95.6
	Does not smoke	1.0 *	2.4 *	96.6
Alcohol drinker	Drinks regularly	0.6 *	2.9 *	96.5
	Drinks occasionally or not at all	1.6 *	2.2 *	96.2
**Total**		**0.97%**	**2.66%**	**96.37%**

Note: * = *p* < 0.05, based on Chi-square tests for significantly different from (1) the reference group with a valid sexual identity response, and (2) the heterosexual reference group among those who answered the question. BMI = body mass index (categories based on World Health Organization cut-offs for risks of impaired health). Source: Canadian Community Health Survey 2008–2017 (n = 202,560) (author’s calculations; proportions bootstrap weighted for population representation).

**Table 2 ijerph-20-05346-t002:** Percentage distribution of survey respondents aged 18–59 by their response to the question on sexual identity, according to the remoteness of the residential community.

Relative Remoteness		(1) Did Not Answer	(2) Answered	
			Lesbian, Gay, or Bisexual	Heterosexual
	Quintile 1-highly accessible areas	1.1 *	2.8 *	96.1
	Quintile 2-accessible	0.7 *	2.6 *	96.7
	Quintile 3-moderately accessible	0.7 *	1.9 *	97.4
	Quintile 4-remote	0.5 *	1.7 *	97.8
	Quintile 5-very remote areas	0.6 *	1.3 *	98.1
**Total**		**0.97%**	**2.66%**	**96.37%**

Note: * = *p* < 0.05, based on Chi-square tests for significantly different from (1) the reference group with a valid sexual identity response, and (2) the heterosexual reference group among those who answered the question. Residential remoteness is based on quintiles of community accessibility and remoteness, with quintile 1 = most urban/accessible areas of the country and quintile 5 = most rural/remote areas. Source: Canadian Community Health Survey 2008–2017 (n = 202,560) linked to the Index of Remoteness for all inhabited communities (author’s calculations; data weighted for population representation).

**Table 3 ijerph-20-05346-t003:** Proportion (%) of survey respondents aged 18–59 with cardiometabolic conditions and healthcare service use, according to their response to the question on sexual identity.

Characteristic		(1) Did Not Answer	(2) Answered		Total
			Lesbian, Gay, or Bisexual	Heterosexual	
Condition	Has diabetes	9.5 *	3.4	3.9	3.91%
	Has hypertension	11.1 *	6.0 *	8.5	8.47%
	Has heart disease	3.0 *	2.2 *	1.8	1.79%
	Suffers effects of stroke	0.9 *	0.4	0.4	0.42%
	Any cardiometabolic condition	17.1 *	9.3 *	11.7	11.69%
Healthcare service use	Has a regular healthcare provider	83.6	79.8 *	84.7	84.53%
	Hospitalized for a cardiometabolic condition	4.9 *	3.7 *	4.9	4.88%

Note: * = *p* < 0.05, based on Chi-square tests for significantly different from (1) the reference group with a valid sexual identity response, and (2) the heterosexual reference group among those who answered the question. Hypertension status is based on self-reports of having been diagnosed with or taking medication for high blood pressure. Hospitalization status is based on having been admitted at least once over the period of observation for diabetes, hypertension, health disease, heart failure, or stroke. Source: Canadian Community Health Survey 2008–2017 linked to Discharge Abstract Database 2008/09–2017/18 (author’s calculations; data weighted for population representation).

## Data Availability

The data that support the findings of the study are available through Statistics Canada’s Research Data Centres, but restrictions apply to the public availability of these confidential data, which were used with permission for the current study. Researchers who meet the eligibility criteria to access the microdata may apply at: www.statcan.gc.ca/en/microdata/data-centres/access (accessed on 16 January 2023).

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
