# Peer review of "Evaluation of Survey Nonresponse in Measuring Cardiometabolic Health Risk Factors and Outcomes among Sexual Minority Populations: A National Data Linkage Analysis"

_ijerph, 2023, doi:10.3390/ijerph20075346_

Round 1

Reviewer 1 Report

This is an interesting study that uses an innovative linking approach to explore the issue of nonresponse to questions about sexual identity in surveys. 

Would the authors expect that participants who consented to have their data shared and linked with certain partner sources differ from those who did not conesnt to data sharing?  If there are expected differences (e.g., those who do not consent to linking data sources may be more concerned about privacy), how would these differences affect the interpretation of the findings from the current analysis?

It is suprising that those who did not report sexual identiy were LESS often residing in rural and remote communities; one might expect that individuals living in rural or remote communities would be more likely to skip such a question due to privacy concerns or living in a more conservative area. 

This study makes an important point that those who prefer not to state their sexual identity should be retained in analyses that focus on sexual/gender minority status and that the motivations for nonresponse should be carefully considered.  It would be helpful if the authors were to make some recommendations with regards to best practices for handling participants who choose not to answer questions regarding sexual identity, if possible. 

Author Response

We thank the reviewer for their appreciation of our study. We have made some revisions to the earlier draft manuscript based on the constructive feedback.

Regarding the point on participants who had not consented to have their data shared and linked with partner sources: we were unable to profile their characteristics by nature of the privacy protocols (i.e., we did not have informed consent to access their data for the present study). We have teased out a separate subsection in the Discussion to better highlight this study limitation, noting that the extent to which the characteristics are different/similar remains unknown. 

We were similarly surprised about the finding linking the likelihood of disclosing one's sexual identity with rural/remote residence. In the Discussion, we have added the qualifier that rural healthcare services should reconsider "perceived" challenges associated with identity disclosure, given the lack of actual evidence on this issue. That said, we also noted among the study limitations that the risk of sampling errors might be heightened for the most rural/remote areas. 

We appreciate the suggestion of offering recommendations for survey improvement. We have added some details in the Conclusion, supported by  evidence-based research elsewhere (given that we did not specifically investigate the validity of using different types of questions to identify LGB persons).

Reviewer 2 Report

The study by Gupta and Cookson on the Evaluation of survey nonresponse in measuring cardiometabolic health risk factors and outcomes among sexual minority populations: a national data linkage analysis is an important and commendable contribution to the field of health disparities. 

However, their results, and coming from a middle-income country where rural and ethnic diversity account for some very pressing and systematic health disparities, makes the reader wonder why they think that excluding First Nations data, more than a study limitation is itself a bias.

Perhaps they could include some information on cardiometabolic health risk factors among these groups considering other health disparities the country has made publicly available. See Richmond, C.A.M., Cook, C. Creating conditions for Canadian aboriginal health equity: the promise of healthy public policy. Public Health Rev 37, 2 (2016). https://doi.org/10.1186/s40985-016-0016-5

Author Response

We thank the reviewer for their appreciation of our manuscript, and for the constructive feedback.

We fully agree that many ethnic minorities experience significant health disparities, within and across countries at different levels of development. Please note that the exclusion of some First Nations reserves from the Canadian Community Health Survey's sampling frame was a methodological path by the national statistical agency (i.e., outside the remit of our research). Other sources capture data regarding on-reserve Indigenous communities (notably, the regional First Nations Health Surveys), but these were outside the scope and privacy protocols of our present work. We have added more details to the Discussion in a "study limitations" subsection to better frame the challenges. We have specified that the inability to disaggregate the data on sexual minority subpopulations by ethnocultural background due to privacy protocols is not unique to our paper (i.e., we have cited another publication using unlinked survey data that raised the same challenge for off-reserve Indigenous persons). We have also added a reference to highlight this important information gap, notably one that specifically addresses Two-Spirit Indigenous health, and some updated information on how the statistical agency is moving forward in this direction.